# Host-Induced Gene Silencing: A Powerful Strategy to Control Diseases of Wheat and Barley

**DOI:** 10.3390/ijms20010206

**Published:** 2019-01-08

**Authors:** Tuo Qi, Jia Guo, Huan Peng, Peng Liu, Zhensheng Kang, Jun Guo

**Affiliations:** State Key Laboratory of Crop Stress Biology for Arid Areas, College of Plant Protection, Northwest A&F University, Yangling 712100, Shaanxi, China; 2014060057@nwsuaf.edu.cn (T.Q.); guojia1889@126.com (J.G.); 18821675862@163.com (H.P.); wood319@126.com (P.L.); kangzs@nwsuaf.edu.cn (Z.K.)

**Keywords:** HIGS, transgene, wheat, barley

## Abstract

Wheat and barley are the most highly produced and consumed grains in the world. Various pathogens—viruses, bacteria, fungi, insect pests, and nematode parasites—are major threats to yield and economic losses. Strategies for the management of disease control mainly depend on resistance or tolerance breeding, chemical control, and biological control. The discoveries of RNA silencing mechanisms provide a transgenic approach for disease management. Host-induced gene silencing (HIGS) employing RNA silencing mechanisms and, specifically, silencing the targets of invading pathogens, has been successfully applied in crop disease prevention. Here, we cover recent studies that indicate that HIGS is a valuable tool to protect wheat and barley from diseases in an environmentally friendly way.

## 1. Introduction 

Cereal grains, including major cereal grains (e.g., wheat and rice) and other minor grains (e.g., barley and oats) have provided over 56% of the caloric and 50% of the protein requirements in human diets for thousands of years, since their domestication. Wheat is one of the most widely grown small-grain cereal crops around the world [1]. About 721 million tons of wheat annually was produced globally from 2012 to 2016, and the record of 760 million tons of global wheat production was reached in 2016, according to the Food and Agricultural Organization (FAO) of the United Nations (http://www.fao.org/worldfoodsituation/csdb/en/, access on 17 October 2018). China, India, the United States of America, the Russian Federation, and France are the main producers in the world and provide the production of half of the world’s wheat. Barley is also one of the most important minor grains. According to the FAO, 142 million tons of barley was produced annually worldwide from 2012 to 2016. The Russian federation, France, Germany, and Australia provide one third of the total production. Barley can be used in beer brewing and is also an important feed for the livestock industry. The yields of the global barley supply were related to beer consumption [2]. Although the development of modern agricultural science and technology greatly reduced the yield loss, an average of 10–15% of the global crop production (more than 300 million tons) is still threatened by plant diseases [3,4]. With the increasing world population, the demand for crop products, combined with food security and balanced nutrition, are rapidly increasing [5]. High-yielding and disease resistant varieties are required at unprecedented levels.

Crop plants are subject to diseases caused by parasitic insects, nematode parasites, pathogenic viruses, bacteria, oomycetes, and fungi. Crop yields and their associated economic losses are major global concerns in modern agriculture. Aphids infesting wheat and barley, including grain aphid (*Sitobion avenae* F.), Russian wheat aphid (*Diuraphis noxia*), and greenbug (*Schizaphis graminum* Rondani), are major agricultural pests in crop plants, not only because of feeding injury, but also because of vectoring viruses (i.e., *Barley yellow dwarf virus* (BYDV) and *Triticum mosaic virus* (TriMV)). Coupled with the presence of parasitic nematodes (*Heterodera* spp.) on wheat and barley, hundreds of millions of dollars are lost every year. In addition, the fungal diseases, including the rusts, blotches, powdery mildew, and head blight/scab, of wheat and barley are currently prominent threats. New virulent races of stem rust fungus, such as strain Ug99 in Africa and the Middle East and V26 in China [6], have caused tremendous yield losses in wheat production. Wheat blast appeared suddenly in 2016 and destroyed wheat fields in Bangladesh [7]. *Fusarium graminearum* ravaged fields of cereal crops and produced mycotoxins that threatened food security. The continual pursuit for yield and quality is a big challenge for breeds, and the constant loss of suitable farm land, unpredictable natural calamity, and the epidemic of multitudinous disease severely hinder the production of wheat and barley. Widely cultivated high-performing varieties lead to disease pandemics and to the loss of genetic diversity. Thus, so many examples remind us that disease resistance breeding and selection are urgently needed.

An array of approaches has been applied to manage crop diseases, such as agrochemical applications, biological control, host resistant selection and breeding, and crop management strategies. The most effective strategy is the use of resistant cultivars combined with reasonable management methods. Because of the rapid emergence of virulent races and fungicide-resistant pathogen strains, traditional breeding is insufficient to combat the abundance of crop diseases. The development of biotechnological approaches provides a novel approach to obtain disease-resistant plants that not only display a high resistance to multiple pathogens, but that are environmentally safe and sound. Based on the knowledge of the molecular patterns involved in plant–microbe interactions, genetically modified plants via transgene-based host-induced gene silencing (HIGS) may be a new effective, environmentally-friendly approach to controlling the crop diseases caused by parasitic pests, nematodes, viruses, and fungi.

## 2. General Mechanism of HIGS

Plants naturally develop an immune system, based on the RNA silencing machinery, to defend against invading viruses [8,9,10]. This feature has been utilized to develop host-induced gene silencing technology (HIGS) to control other plant pathogens [11]. HIGS is a further development of virus induced gene silencing (VIGS) [9], which allows for the silencing of genes in plant pathogens by expressing an RNA interference (RNAi) construct against specific genes endogenous to the pathogen in the host plant. HIGS is an RNAi-based process where small RNAs made in the plant silence the genes of the pests or pathogens that attack the plant. The small RNAs are generally made by producing double stranded RNA (dsRNA) in transgenic plants, but for experimental purposes, the dsRNA can be introduced into the plant cells with agrobacterium or viruses that replicate through dsRNA. RNAi, discovered in the expression of a chimeric chalcone synthase gene in petunia, led to a reversible homologous gene suppression [12]. RNA silencing is a highly-conserved mechanism that operates in most eukaryotes, including plants, animals, and fungi. Andrew Z. Fire and Craig C. Mello received the Nobel Prize in Physiology or Medicine in 2006 for their contribution to RNA interference [13]. One of the major features of RNA silencing is the production of small RNAs of 21–30 nucleotides (nt) in length, which can regulate gene expression in a sequence-specific manner. Small RNAs generated from dsRNA guide transcriptional gene silencing (TGS) and post-transcriptional gene silencing (PTGS). PTGS involves the recognition and silencing of mRNA in the cytoplasm, whereas TGS involves RNA-mediated DNA methylation in the promoter region, which suppresses the specific gene expression. Typically, Dicer or Dicer-like (DCL) proteins recognize a double-stranded RNA and process it into smaller RNAs. These sequence-specific small RNAs are hired by Argonaute (AGO) protein and are processed into an RNA-induced multi-subunit silencing complex (RISC). The RISC regulates the target genes to achieve gene silencing in a sequence-specific manner. 

With the deep insight of the RNAi mechanism, researchers found that RNAi can be triggered by the dsRNA or DNA from a virus in higher plants, the foreign dsRNA that can be silenced endogenously is a natural immunity to defense against virus [9], the specialty of RNAi that it has sequence specificity and is triggered by introducing exogenous dsRNA makes RNAi a perfect tool to analyze the genomic function in the parasitic system. VIGS is based on the RNAi-mediated natural immunity to defense against viruses, by importing the foreign gene in the viral vector. The initial long RNA precursors at the target sites, transcribed by RNA polymerase IV, and as a template for RNA-dependent RNA polymerase 2 (RDR2) to produce dsRNA, dsRNA is processed into 24 nt hetsiRNAs by DCL3. HetsiRNAs, which represent secondary siRNAs (hetsiRNAs), are the most abundant class of siRNAs in plants. HetsiRNAs establish and maintain silencing through the epigenetic pathway. Endogenous expressing siRNA in plants always need TGS or RNA-directed DNA methylation (RdDM), which is an epigenetic regulation essential for silencing. dsRNA is also processed by DCL2 and DCL4 to 21–22 nt siRNAs. The siRNAs induced cleavage of target mRNA can trigger the production of secondary siRNAs (secsiRNAs). Most secsiRNAs are related to PTGS and the spreading of RNA silencing locally or systemically. SecsiRNAs is important for the amplification of RNA silencing in plants, especially the VIGS and HIGS. HIGS is a transgenic modification technology that relies on the instruction of an inverted repeat sequence into the plant genome. The sequence homolog to the gene from pathogens integrated into the plant genome express the siRNA targeted gene and result in the gene silencing of pathogens. Various HIGS vectors have been constructed to drive the long dsRNA as well as hairpin sequence through an inverted repeated gene or an inverted promotor sequences in to the plant genome. Foreign dsRNA or hairpin sequence are introduced into the plant genome and are cleaved by endonuclease-type DICER enzymes into siRNAs, with the amplification of the hetsiRNAs and secsiRNAs, and these siRNAs are transported systemically and taken up by the fungal cells or other parasites (Figure 1) to silence the fungal gene, which is critical for the invasion [14]. In higher plants, RNAi not only regulates the endogenous gene expression, but the instruction of foreign dsRNA in plants has also recently been a conventional agriculture plant protection.

Common wheat (*Triticum aestivum*) is an AABBDD hexaploid (AA from *Triticum uratu*, BB from *Aegilops* species and DD from *Triticum tauschii*) with a complex genome containing approximately 17 billion (17 Gb) nucleotides [15]. Barley (*Hordeum vulgare* L.) is a diploid monocotyledon with seven chromosomes containing about 5.1 billion (5.1 Gb) nucleotides [16]. The huge and complex genomes of wheat and barley complicate the isolation and functional description of the individual genes. HIGS is an effective and convenient tool to explore the molecular genetics of disease resistance in crops, and a series of studies have been performed in wheat and barley using this technology. For wheat and barley, BSMV-induced gene silencing enables the analysis of large-scale functional genomics [17]. Moreover, transgene-induced RNAi could be a useful tool for genetically modified plants to achieve high-value crops [18]. Basically, trigger selection is the primary concern for RNAi. The trigger size and sequence identity between the trigger and its target genes must be considered before RNAi experiments can be designed [19,20,21]. Trigger sizes from 200–550-bp have been successfully used in wheat [22]. A perfect identity of the trigger and its targets is necessary for a continuous stretch of siRNA [23,24,25]. Reducing the off-target effects is another factor that must be considered. Bioinformatics analysis, such as Blast and SIFI software, are of considerable value. Overall, the RNAi modification of wheat and barley genes is an environmentally friendly and effective strategy to engineer novel phenotypes, and the perfect parameters suitable for this system will be amended with caution in the future.

## 3. HIGS Used for Controlling Diseases of Wheat and Barely

Wheat and barley are susceptible to various biotic stresses, such as grain aphid (*Sitobion avenae* F.), Russian wheat aphid (*Diuraphis noxia*), greenbug (*Schizaphis graminum* Rondani), *Barley yellow dwarf virus* (BYDV), *Triticum mosaic virus* (TriMV), parasitic nematodes (*Heterodera* spp., *Tylenchus tritici*), and bacterial pathogens (*Bacterium tritici*). Fungal pathogens are major threats to wheat and barley, causing huge economic losses. Take-all, caused by *Gaeumannomyces graminis* var. *tritici*, is a serious disease of barley and wheat. Powdery mildew, caused by *Blumeria graminis*, also devastates both hosts. Fusarium head blight, also known as scab, caused by *Fusarium* species and rust fungi (*Puccinia species*), causes disastrous yield and quality losses in cereals. Other pathogens, such as *Zymoseptoria tritici*, *Parastagonospora nodorum*, *Pyricularia graminis-tritici*, and *Pyrenophora tritici-repentis*, threaten crop production and food safety. The success of traditional disease management strategies (e.g., irrigation management, chemical control, and agronomic practices) are largely influenced by the environment, whereas RNAi-based HIGS technology provides a novel and innovative approach to control crop diseases. Here, we summarize the recent studies of HIGS in wheat and barley (Table 1), with consideration of the key diseases affecting crop production, and briefly address the current status of the RNAi system used in gaining an understanding of the molecular mechanism of each interaction.

### 3.1. HIGS against Insect Pests

The expression of siRNA in transgenic plants provides protection to wheat, barley, and other small grains against individual insect species by targeting the genes in aphids and other parasitic pests that threaten crops. Transgenic siRNA-expressing plants like maize and corn have proven to be effective in insect control. For example, *Bacillus thuringiensis* (Bt) has been successfully used for the management of cotton insect pests [40]. Four targets from 66 unigenes were isolated for RNAi against grain aphids in wheat [41]. In addition, five potential RNAi targets have been identified from 5490 unigenes of grain aphids in wheat [42]. Several studies have confirmed that heat-shock protein 90 (HSP90) is an ideal target, because of its crucial role, ubiquity, and conservation [43,44]. The silencing of the chitin synthase gene prevented the insects from hatching [45], and from knocking down the segmentation gene, hairy, which prevented insect feeding [46]. The silencing of the essential genes (e.g., ecdysone receptor (EcR) and ultraspiracle protein (USP)) in grain aphids (*Sitobion avenae* F.) reduced the survival and fecundity of wheat, providing a persistent and transgenerational method for improving wheat resistance [47]. Silencing the matrix metalloproteinase MMP-2 is lethal, because of its importance in gut development [48]. Additionally, several potential RNAi targets were confirmed through feeding or injection in the grain aphid *Sitobion avenae*, such as genes encoding catalase, acetylcholinesterase1, cytochrome c oxidase subunit VIIc precursor, zinc finger protein, secreted salivary peptide DSR32, salivary protein DSR33, serine protease 1 DSR48, and olfactory co-receptor [49]. Not only can individual genes be targeted by RNAi, their promoters are perfect elements for controlling gene expression, and, therefore, can be used in the RNAi strategy. Replacing the promoter of the HIGS constructs to the *SUC2* expressed along the sieve tubes to the transport solutes can help control tissue-specific feeding insect pests [50]. The first genomic modification of plants against pests was produced in 1983 [51]. Transgenic wheat expressing dsRNA to knock down the CarE gene (*CbE E4*) of *Sitobion avenae* delayed larval growth and reduced resistance to pesticides [26]. The transgenic expression of siRNA of the structural sheath protein (SHP), which is an essential component of the leaf sheath in barley, strongly reduced the feeding and spread of aphids [27,52], and the effects were sustained for seven generations [52]. Coupled with the dsRNA in transgenic plants, which are also heritable, the control of crop diseases would be continuous. HIGS for controlling the insect pests of crop plants is becoming an environmentally friendly and convenient commercial product.

### 3.2. HIGS against Nematodes

The damage caused by nematodes hinges mainly on the migratory or sedentary phases of this species. The exo-parasitic species live in the soil and use long stylets to feed on epidermal cells; the endo-parasitic, mobile nematodes live inside the plant tissue and feed on non-lignified cells; other species are the sedentary nematodes, which are effectively managed through RNAi [53]. The yield losses of wheat in the presence of nematodes are caused mainly by cyst nematodes (*Heterodera* spp.), which also threaten the production of barley [54]. RNA interference in several nematode species (e.g., cyst nematodes, root knot nematodes, root lesion nematodes, and other ectoparasitic nematodes) has been explored through feeding, soaking, injection, or in planta delivery [28,29,55]. The knockdown of the expression of *pat*-*10* and *unc*-*87* of *Pratylenchus thornei*, which attacks the wheat roots, reduces reproduction by 77–81% [29]. Cysteine proteinase, C-type lectin, major sperm protein, chitin synthase, aminopeptidase, β-1,4-endoglucanase, secreted amphid protein, FMRF amide-like peptides (flp-14 and flp-18), pectate lyase, chorismate mutase, secreted peptide SYV46, dual oxidase, splicing factor, integrase, and secreted peptide 16D 10 were reviewed as potential RNAi targets by Lilley et al. (2007) [28]. The phenotypic effects of these RNA interference experiments were commonly a reduced number of established nematodes or an increase in the male population, which indicates that juveniles experience adverse conditions [28]. The first successful application of HIGS in 2006 was used to confer nematode resistance by protecting the host from infection [56]. Several factors in nematodes, such as Cpn-1, Y25, Prp-17, tyrosine phosphatase, calreticuline Mi-CRT, a parasitism gene 8D05, ribosome 3a, 4, synaptobrevin, and a spliceosomal SR protein have been shown to be good candidates for HIGS for improving resistance to nematodes [57]. These reports provide the foundation and confidence to control nematodes through RNAi strategies.

### 3.3. HIGS against Viruses

To date, no technologies are available to cure virus-infected plants, and the only way to limit the spread of viruses is via the application of costly chemical treatments eliminating the virus-transmitting organisms. The selection of virus-resistant cultivars has been the most effective solution, but natural resources of resistance are not sufficient. In the past few years, HIGS against viruses has proven to be an efficient technology. *Wheat streak mosaic virus* (WSMV) infection is inhibited using the full-length sequence of the viral replicase (Nib) in transgenic wheat. Interestingly, only one of six lines can detect the transgene mRNA, which did not provide resistance to the virus [58], suggesting that post-transcriptional gene silencing was involved in the transgenic wheat resistance. Furthermore, the full length of the WSMV coat protein gene transformed into wheat, conferred resistance to WSMV by two groups, respectively [59,60]. Li and co-authors (2014) found a strong resistance phenotype in some transgene lines for the first generation, but the T2 and T3 generations lost the phenotype. In turn, the use of hairpin-forming transgenes of the WSMV CP protein elicits efficient resistance. This is also the first report of stable WSMV-resistant wheat, which has stable resistance lines through the T5 generation [32]. The second report of the stable transgene (T6 generation) of RNAi-mediated resistance in wheat, and also the first report of the *Triticum mosaic virus* (TriMV), was accomplished by constructing a hairpin sequence of the TriMV CP gene [61]. Polycistronic artificial miRNA-mediated resistance was also demonstrated as an efficient tool. The artificial microRNAs (amiRNAs) can be designed by targeting the different conserved sequence elements of the viruses. Potential amiRNA sequences were selected to minimize the off-target effects. Fahim and colleagues (2016) concluded that polycistronic amiRNAs can be utilized to induce virus resistance [30]. In addition, amiRNA technology was also used for the introduction of highly efficient resistance in barley against *Wheat dwarf virus* (WDV), a DNA virus belonging to the *Geminiviridae* family, and the resistance is effective at low temperatures [31]. 

### 3.4. HIGS against Fungi

Fungal pathogens cause more than 70% of the yield loss of crops worldwide, and RNAi strategies have been widely used to characterize the gene function and to construct transgenic lines against the fungal pathogens of wheat and barley. Resistance genes are not always useful for some fungi (e.g., *Fusarium* species and *Puccinia* species), because of the associated toxins and rapid evolution of virulent strains. RNAi-based HIGS provide a novel, innovative approach to control the crop diseases caused by fungi. Here, we review the major diseases [62] that threaten the production of small grains.

The powdery mildew of barley caused by *Blumeria graminis* f. sp. *hordei* and wheat caused by *B. graminis* f. sp. *tritici* are serious diseases. Nowara et al. (2010) showed that dsRNA targeting the avirulence gene *Avra10*, which is recognized by the resistance gene *Mla10*, significantly reduced fungal development in the absence of *Mla10*, and the silencing of 1,3-b-glucanosyltransferases (*BgGTF1* and *BgGTF2*) reduced the early development of the pathogen [33]. Pliego et al. (2013) screened eight targets (e.g., β-1,3 glucosyltransferases, metallo-proteases, microbial secreted ribonucleases, and proteins of a known function) from fifty *Blumeria* effector candidates, that contribute to fungal development through the transient expressing inverted repeat sequence of these gene of effectors into barley leaves [34].

Rust disease, caused by *Puccinia* species, which are critical obligate biotrophic organisms [63], threaten the world’s crop production and have caused enormous losses ranging from 4.3 to 5 billion US dollars annually, especially for wheat and barley [64]. Yin et al. (2010) [65] established a *Barley stripe mosaic virus* (BSMV)-induced gene silencing system to knockdown *Puccinia striiformis* f. sp. *tritici* (*Pst*) genes. BSMV-HIGS provide a way to analyze the function and to screen RNAi targets for the control of rust diseases. Panwar et al. [66] transiently silenced the genes encoding mitogen-activated protein kinase 1 (*PtMAPK1*), cyclophilin (*PtCYC1*), and calcineurin B (*PtCNB*) from *Puccinia triticina* (*Pt*) through *A. tumefaciens*-mediated transformation, and the disease symptom significantly changed and the sporulation decreased macroscopically. The fungal biomass and emergence of uredinia also declared that silencing *PtMAPK1*, *PtCYC1*, and *PtCNB* reduced the virulence and restricted the development of rust fungi, including *Puccinia graminis* and *Puccinia striiformis*. Many factors in the *Puccinia* species have been successfully silenced by BSMV-HIGS, such as protein kinase (*PsSRPKL*) [67], small GTP-binding protein (*PsRan*) [68], distinct *Ras* genes (*PsRas1* and *PsRas2*) [69], MADX-box transcription factor (*PstMCM1-1*) [70], transcription factor *PstSTE12* [71], MAP kinase kinase kinase *PsKPP4* [72], secreted protein gene *Pst_8713* [73], effector (*PSTHa5a23*) [74], effector *PEC6* [75], and Zn-only superoxide dismutase (*PsSOD1*) [76]. Most of these genes (e.g., *PsSOD1*, *PsSRPKL*, and *PsRas2*) contribute to the virulence, and the knockdown them results in a substantial restriction in the growth and spread of the hyphae of *Pst*, the portion of these genes (e.g., *PsRan* and *PsSRPKL*) silenced through BSMV-HIGS leading to the morphological abnormalities, such as the hyphal expansion or haustorium shrink of rust fungi. Qi et al. (2018) produced siRNA targeting the gene of the catalytic subunit of protein kinase A (*PsCPK1*) in transgenic wheat and observed resistance against *Pst* through the fourth generation, and showed that the cAMP signaling pathway genes were down-regulated [35]. Additionally, Zhu et al. (2017) produced transgenic wheat lines expressing a RNAi construct against the MAP kinase kinase gene (*PsFUZ7*), and proved them to be significantly resistant to *Pst.* Infection by the virulent race was strongly restricted, and only rarely produced urediniospores or caused substantial necrosis on transgenic wheat leaves [14]. The transgenic wheat, which stably expressed the hairpin RNAi constructs of the homologous gene of MAP kinase (*PtMAPK1*) and the cyclophilin (*PtCYC1*) of leaf rust, restricted the fungal development and dramatically reduced the fungal biomass [36]. 

Wheat scab (also called Fusarium head blight) is a devastating disease mainly caused by *F. graminearum*. HIGS technology has also been used to control this disease. Koch et al. (2013) constructed dsRNA in barley expressing the siRNA targeting fungal sterol cytochrome P450 lanosterol C-14a-demethylase gene (*CYP51*), and found that it restricted fungal growth and alleviated disease symptoms. Furthermore, the RNAi transgenic barley and *Arabidopsis* containing three constructs (*CYP51B*, *CYP51A*, and *CYP51C*) that target the three paralogs of CYP51 are completely immune against *F. graminearum* [37]. The RNA interference of the chitin synthase (Chs) gene conferred stable resistance for five generations and reduced the deoxynivalenol (DON) production when three hairpin RNAs of *Chs* were transferred into two wheat cultivars (Yangmai15 and Sumai3) [38]. Chen et al. (2016) stably expressed the dsRNA of β-1,3-glucan synthase gene *FcGls1* in wheat against *F. culmorum* and found that it enhanced the Fusarium head blight (FHB) resistance in the leaf and spike [39]. Machado et al. (2018) covered the recent advances in the RNAi-based methods in the control of FHB, and highlighted that despite some of these disadvantages of the RNAi approach, HIGS has emerged as a promising new approach to control fungal plant diseases [77]. Wegulo et al. (2015) reviewed strategies, including planting resistant or tolerant cultivars, spraying chemicals, improving cultivation techniques, and harvesting strategies to manage FHB and reduce the toxin content, and the author declared that biological control agents may be the most effective application, and that new strategies and new sources of resistance introgression into new cultivars need to be developed for the management of FHB and DON more effectively in the future [78], and that HIGS technology would be a novel and effective way to control head blight—the use of HIGS on a commercial scale appears possible in the near future.

These studies provide a proof-of-concept that HIGS is an effective strategy to control the fungal diseases of small grains, although perfect targets and conditions still must be determined in order to produce resistance in wheat and barley, especially to wheat blast (WB) caused by Magnaporthe oryzae Triticum (MoT); take-all caused by *Gaeumannomyces graminis* var. *tritici*; and blotch diseases caused by Zymoseptoria tritici, Parastagonospora nodorum, and Pyrenophora tritici-repentis. Breeding for and sustaining multiple disease resistance against complex and mixed disease situations is a challenging task.

## 4. Current Challenges of HIGS

HIGS technology as a novel plant genomic modification tool that has been used to improve resistance against various diseases, breed high quality crops, develop seedless plant varieties, eliminate fungal toxins, improve stress tolerance, and develop male sterility varieties for breeding [18]. Significant challenges must be overcome before RNAi-mediated transgenic plants can be deployed in the field. Parasites and pathogens deliver small RNAs into their host to modify the host defense responses. Small RNAs also can be transferred into pathogens and pests to depress the invasion through feeding on plants and parasitism [79]. Researchers have proved that siRNAs and dsRNA can be transferred into fungal cells efficiently. External incubation with siRNA leads to the down regulation of a specific gene in *Aspergillus*, which proved the absorption of siRNA [80], and the silencing of the specific fungal gene was also interfered by the external treatment of siRNAs and dsRNAs in *Botrytis cinerea* [81]. However, it is still shrouded in mist whether and how siRNAs and/or dsRNAs are transported, and whether the intact RNAi machineries of the parasite are necessary or not. Recent studies report that exogenous plant miRNAs are sufficient to regulate animal target genes by feeding on the transgenic plants [82]. These findings alter the vistas for the safety of transgenic plants (not only the target gene, but also the selection makers), and should be interpreted with caution. The methods of selecting an appropriate target and the most suitable fragment of HIGS to delay or even entirely restrict disease still need to be defined. The efficiency of HIGS relies on enough supplements and on the successful transportation of siRNA between the two organisms. Accordingly, HIGS cannot be used against necrotrophic fungi, as they absorb nutrition and other metabolites from dead host tissue, which could not supply sufficient amounts of siRNAs. In addition, silencing an individual gene of the pathogens may not be sufficient to control disease, because of functional redundancy, and the incomplete silencing of mRNA levels does not guarantee the deactivation of the protein. This drawback can be overcome by using the transient silencing system for high-throughput screening. Off-target effects are also serious problems when we construct transgenic plants and try to avoid the off-target influence of unwanted symptoms or effects on agronomic traits. Bioinformatic tools (e.g., SIFI and Blastn) are very useful. In addition, HIGS is not always available in certain tissues (e.g., root and fruit) or systems (e.g., soil-borne fungal pathogens and insect groups that block siRNA uptake), and many crop species are not suitable for genomic transformation [83,84]. Although recent studies have attempted to express dsRNA in chloroplasts or other plastids so as to avoid the disadvantage of expressing dsRNA in the nucleus, the results show that it is obvious that the artificial constructs would be limited to being spread with the absence of chloroplasts when there is random mating in natural conditions [84]. The development of chloroplast transformation protocols for cereals (e.g., rice, wheat, maize, and barley) will be more commercially and environmentally attractive for plant protection against certain pests and pathogens. 

A spray-induced gene silencing (SIGS) strategy is another innovative RNAi-based disease control strategy, which has been successfully used on both monocots and dicots, against fungi and pest infection. Koch et al. (2016) silenced the three fungal genes encoding cytochrome P450 lanosterol C-14α-demethylases in *F. graminearum* through the RNA spraying method on barley, and successfully restricted fungal development [37,85]. There may be some new theory in the siRNAs transportation. During HIGS against insects, systemic RNA interference deficient (SID) in *Caenorhabditis elegans* functions as a dsRNA transport protein and binds to the specific long dsRNA [86], while homologous genes or similar proteins responsible for the RNA transporters in fungi are not reported yet. While how RNAs are transported from the host tissue into the parasite, especially fungi, are unclear, speculation exists that siRNAs are transferred via specific transporters or some extracellular vesicles [85]. Li et al. (2015) mentioned that fungi produced vesicles to transfer siRNAs to be taken up into plant cells [87]. Recently, the necrotrophic pathogen, plant pathogenic bacteria, and parasitic nematodes delivered vesicles that contain siRNAs into the host tissue and modulated host immunity [88,89,90]. The vesicles are highly related to the RNA communication between the organism and pathogens. However, there is still lack evidence that plant-derived vesicles transmit RNAs to pathogenic fungi. 

Every organism has its specific and stable genome for genetic information storage and for transfer to future generations. However, how does the genome of the organism remain stable, inheritable, and variable for millions of years? The discovery and research of RNA silencing provides a new perspective on this issue. RNAi-derived silencing can inhibit the invasion of foreign DNA (virus and transgenic ways). Maintaining a low activity of endogenous transposons and repeats is an important immune mechanism of gene expression during plant development, while the regulatory elements and regulation directing the signal pathway have not been found. How does the plant endogenous system identify itself and the “exotic” DNA molecules, and consequently the blocking and silencing? This still needs to be explored. Under the supervision of the endogenous RdDM silent epigenetic regulation, the over-expression function difficultly and often produces co-suppression. Conversely, RNAi construction for knocking down the function of endogenous genes can only be made at the site of initiation, and is difficult to pass to the surrounding tissues. The systems used to study the gene silencing pathways are often associated with transgene technology and need to be rethought and verified by future generations. With deeper insights into the mechanism of RNAi, the HIGS strategy of disease control opens novel avenues to improve crop yields.

## 5. Conclusions and Future Prospects

HIGS is a powerful and effective tool in gaining disease resistant transgenic plants and for the functional characterization of vital genes. In the near future, the major question for HIGS strategy will be answered. New targets and fragment selection methods, highly efficient transformation constructs, stable transgenic systems, and other new technologies will enhance the RNAi-derived strategy to develop durable disease resistant plants. RNAi-based technology provides a precise avenue to select high quality traits of varieties. The HIGS strategy has proven to be a novel approach in defending abiotic or biotic stress and quality improvement in an eco-friendly and sustainable manner. Recently, genome editing technologies have progressed rapidly and, in particular, the application of clustered regularly interspaced short palindrome repeats (CRISPR)/CRISPR-associated protein 9 (Cas9) editing has become a powerful tool for the enhancement of pathogen resistance in model plants and important crops [91]. In the near future, through the combined application of HIGS and CRISPR/Cas9, it will become much easier to achieve durable control of the diseases of wheat and barley.

## Figures and Tables

**Figure 1 ijms-20-00206-f001:**
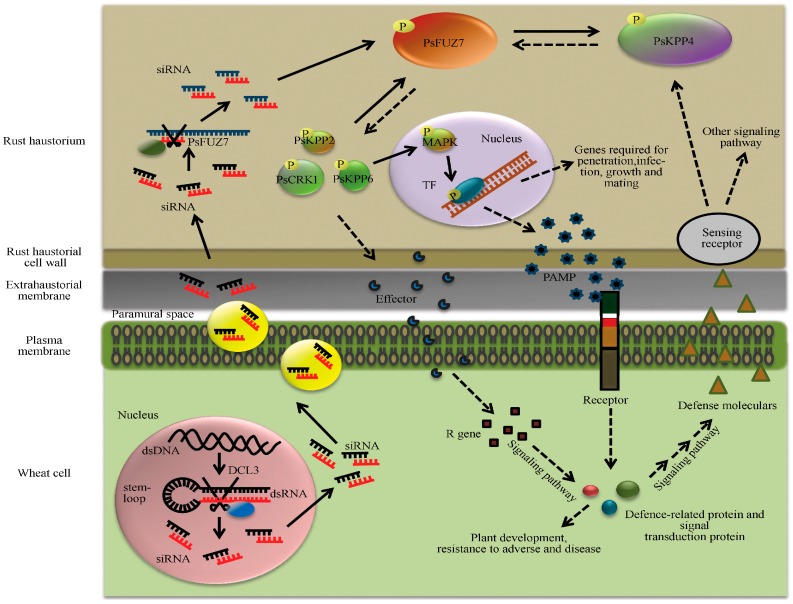
Schematic presentation of possible molecular dialogues between transgenic lines carrying RNAi constructs and colonizing *Pst.* Fungal dsRNA, produced inside transgenic wheat cells, is cleaved by the plant silencing machinery using endonuclease-type dicer enzymes into small silencing molecules (siRNAs). These siRNAs are trapped by a complex of proteins, and are transported to the paramural space. After passing the haustorial cell wall, the silencing molecules trigger the RNAi of their mRNA targets, and may act as primers leading to the activation of systemic silencing signals, thus inducing the immune system of transgenic wheat by mechanisms, including PAMP triggered immunity and Effector triggered immunity [14].

**Table 1 ijms-20-00206-t001:** Summary of study test in wheat and barley by using host-induced gene silencing.

Species		Host	Target Gene	Remark	Major Phenotype	Ref.
**Insects**	*Sitobion avenae*	Wheat	*CbE E4*	Carboxylesterase gene	Impaired tolerance of phoxim insecticides	[26]
	*Sitobion avenae*	Barley	*shp*	Structural sheath protein	Reduced fecundity and inhibited feeding behavior	[27]
**Nematodes**	*Meloidogyne incognita*	Wheat	*HSP90*, *ICL*, and *Mi-cpl-1*	Heat-shock protein 90, isocitrate lyase, and Mi-cpl-1	Reduced reproduction	[28]
	*Pratylenchus* spp.	Wheat	*pat-10*, *unc-87*	Troponin C (Pat-10) and Calponin (unc-87)	Reduced reproduction	[29]
**Viruses**	*Wheat streak mosaic* virus (WSMV)	Wheat	pre-miR395	Artificial microRNA (amiRNA)	Stable resistance	[30]
	*Wheat dwarf* virus (WDV)	Barley	amiR1, amiR6, and amiR8	amiRNAs	Highly efficient resistance	[31]
	WSMV	Wheat	coat protein gene	Coat protein	Consistent resistance	[32]
**Fungi**	*Blumeria graminis*	Wheat, Barley	*Avra10*	Virulence effector	Reduced virulence	[33]
	*Blumeria graminis*	Wheat, Barley	*BEC1011*, *BEC1054*, *BEC1038*, *BEC1016*, *BEC1005*, *BEC1019*, *BEC1040*, and *BEC1018*	Effectors	Reduced virulence	[34]
	*Puccinia striiformis* f. sp. *tritici*	Wheat	*PsCPK1*	PKA catalytic subunit	Stable resistance	[35]
	*Puccinia striiformis* f. sp. *tritici*	Wheat	*PsFuz7*	MAP kinase kinase	Stable resistance	[14]
	*Puccinia triticina*	Wheat	*PtMAPK1* and *PtCYC1*	MAP kinase, cyclophilin	Enhanced resistance	[36]
	*Fusarium* species	Barley	*CYP51*	Cytochrome P450 lanosterol C14-demethylase	Enhanced resistance	[37]
	*Fusarium graminearum*	Wheat	*Chs3b*	Chitin synthase 3b	Stable resistance	[38]
	*Fusarium culmorum*	Wheat	*FcGls1*	β-1, 3-glucan synthase	Enhanced resistance	[39]

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
