# Peer review of "Host-Induced Gene Silencing: A Powerful Strategy to Control Diseases of Wheat and Barley"

_ijms, 2019, doi:10.3390/ijms20010206_

Reviewer 1 Report

The manuscript is a summary of the main examples of host-induced gene silencing against several biotic stresses. The manuscript is written well, and it is efficiently organized. I would spend some more words on the definition of HIGS at line 71, just to better present the technology to readers unfamiliar with it. Also, at the end of the manuscript, I would mention the CRISPR-CAS9 technology and make connections with HIGS.

Author Response

Re: ijms-417210

Host-induced gene silencing: A powerful strategy to control diseases of wheat and barley

****************************************************************************************

Reviewer 1 :

The manuscript is a summary of the main examples of host-induced gene silencing against several biotic stresses. The manuscript is written well, and it is efficiently organized. I would spend some more words on the definition of HIGS at line 71, just to better present the technology to readers unfamiliar with it. Also, at the end of the manuscript, I would mention the CRISPR-CAS9 technology and make connections with HIGS.

Response: According to the reviewer’s suggestions, we have rephrased the manuscript.

Reviewer 2 Report

Weaknesses:

Some of the supporting information could be made more concise, such as the introduction description of the importance of wheat and barley and that those crops are affected by diseases.

Lists, such as lines 151-162 could be turned into tables.

Methods used to test RNAi could be discussed in a new section, in addition to the general mechanism.

Specific points:

Line 91: “analysis” should be “analyze”

Lines 107-109: Font is not consistent. This issue is present throughout the manuscript.

Line 129: “Agilops” is misspelled

Line 192: Reword “Replace”

Line 242: change “coworkers” to “coauthors” or some other term

Line 319: change “harpin” to “hairpin”

Line 402: “millions years” should be “millions of years”

Line 408-409: Reword this sentence

Lines 416-424: The conclusion needs to be revised heavily.

Line 462: Remove “14. ”

Author Response

Re: ijms-417210

Host-induced gene silencing: A powerful strategy to control diseases of wheat and barley

****************************************************************************************

Reviewer 2 :

Some of the supporting information could be made more concise, such as the introduction description of the importance of wheat and barley and that those crops are affected by diseases.

Lists, such as lines 151-162 could be turned into tables.

Methods used to test RNAi could be discussed in a new section, in addition to the general mechanism.

Response: According to the reviewer’s suggestions, we try to rephrase the manuscript.

Specific points:

Line 91: “analysis” should be “analyze”

Response: Revised.

Lines 107-109: Font is not consistent. This issue is present throughout the manuscript.

Response: Revised.

Line 129: “Agilops” is misspelled

Response: Revised.

Line 192: Reword “Replace”

Response: Revised.

Line 242: change “coworkers” to “coauthors” or some other term

Response: Revised.

Line 319: change “harpin” to “hairpin”

Response: Revised.

Line 402: “millions years” should be “millions of years”

Response: Revised.

Line 408-409: Reword this sentence

Response: Revised.

Lines 416-424: The conclusion needs to be revised heavily.

Response: Revised.

Line 462: Remove “14. ”

Response: Revised.